

# A high resolution lithospheric magnetic field model over southern Africa based on a joint inversion of CHAMP, Swarm, WDMAM and ground magnetic field data

Foteini Vervelidou[1], Erwan Thébault[2], and Monika Korte[1]

[1]Helmholtz Centre Potsdam-GFZ German Research Centre for Geosciences, Section 2.3 Geomagnetism, Telegrafenberg, 14473 Potsdam, Germany
[2]Université de Nantes, Laboratoire de Planetologie et Geodynamique, 44322 Nantes Cedex 3, France

**Correspondence:** Foteini Vervelidou (foteini@gfz-potsdam.de)

**Abstract.** We derive a lithospheric magnetic field model up to equivalent Spherical Harmonic degree 1000 over southern Africa. We rely on a joint inversion of satellite, near-surface and ground magnetic field data. The input data set consists of magnetic field vector measurements from the CHAMP satellite, across-track magnetic field differences from the Swarm mission, the World Digital Magnetic Anomaly Map and magnetic field measurements from repeat stations and three local INTERMAGNET observatories. For the inversion scheme, we use the Revised-Spherical Cap Harmonic Analysis (R-SCHA), a regional analysis technique able to deal with magnetic field measurements obtained at different altitudes. The model is carefully assessed and displayed at different altitudes and its spectral content is compared to high resolution global lithospheric field models. By comparing the shape of its spectrum to a statistical power spectrum of Earth's lithospheric magnetic field, we infer the mean magnetic thickness and the mean magnetization over southern Africa.

## 1 Introduction

Thanks to the increasing availability of magnetic field measurements, both at satellite altitude and near or at the Earth's surface, models of the lithospheric magnetic field continuously improve both in terms of accuracy and spatial resolution and cover an increasing range of spatial scales. Marine and aeromagnetic measurements, thanks to their proximity to the lithospheric sources, are capable of capturing well its small scale features. However, its large scale contributions are not accessible by means of these measurements because of their limited spatial extent. Low Earth Orbiting magnetic satellite missions, such as CHAMP (Reigber et al. (2002)) and the ongoing multi-satellite mission Swarm (Friis-Christensen et al. (2006), Olsen et al. (2013)), complement near-surface measurements by offering a consistent global view of the large scale features of the lithospheric magnetic field. The MF7 model, for example, based on CHAMP satellite measurements, is derived up to Spherical Harmonic (SH) degree 133 (Maus et al. (2008)) and the LCS-1 model, based on a joint consideration of CHAMP and Swarm measurements, pushes this limit to SH degree 185, although with a significant loss of energy beyond SH degree 133 (Olsen et al. (2017)).

Thanks to the availability of satellite measurements and to a coordinated international effort to collect and merge all publicly available near-surface measurements, the first global grid of lithospheric field anomalies, the World Digital Magnetic





Anomaly Map (WDMAM) was published in 2007 (Khorhonen et al. (2007)), by the WDMAM Task Force, under the auspices of the International Association of Geomagnetism and Aeronomy (IAGA) and the Commission for the Geological Map of the World (CGMW). The MF5 model (Maus et al. (2007)), based on CHAMP measurements was used to account for its long wavelengths. Based on a different way of processing and merging the marine and aeromagnetic measurements, Maus et al.

(2009) derived another anomaly grid, the EMAG2, which was converted by Maus (2010) to a SH model up to degree 720 (see https://www.ngdc.noaa.gov/geomag/EMM/ for an updated version, the EMM2017 model). In 2015, the second version of the WDMAM was published (Lesur et al. (2016), www.wdmam.org). New data sets were added and the data gaps over the oceans were handled in a different way (see Dyment et al. (2015)). The second version of the WDMAM has been converted to a SH model up to degree 800 (henceforth termed WDMAMv2-SH800).

These global models represent the first attempt to merge the information content of satellite and near-surface measurements in the form of SH models. However, this merging is not carried out by a joint inversion of satellite and near-surface measurements, mainly because of the existence of wide data gaps that lead to numerical instabilities. The high inhomogeneity in data availability calls for regional modeling approaches that are flexible in adapting their spatial resolution to the available data coverage. Over regions well covered by magnetic field measurements, regional models can achieve higher accuracy and

spatial resolution than global models. A review of available regional modelling techniques in the framework of geomagnetism is given by Schott and Thébault (2011). More recent regional techniques to jointly model data at different altitudes include the vector slepian functions (Plattner and Simons (2014), Plattner and Simons (2017)) and the use of equivalent potential field sources (Kother et al. (2015), Kother (2017)). Once lithospheric field models are derived, they can be analyzed in the spatial and spectral domains in order to infer properties of the underlying sources, like the Curie depth and the magnetization (e.g.,

Whaler (2003), Ravat et al. (2007); Bouligand et al. (2009); Rajaram et al. (2009); Scheiber-Enslin et al. (2014), Vervelidou and Thébault (2015)).

In this study, we focus on the southern part of Africa, which is well covered by near surface measurements, and we opt for the Revised Spherical Cap Harmonic Analysis (R-SCHA, Thébault et al. (2006)). We use our model to infer the thickness of the magnetized layer, henceforth termed the magnetic thickness, and the magnetization intensity. In section 2 of this paper we

present the input dataset. In section 3, we describe the modeling method. Then, we present the model with plots at various altitudes, and assess it by showing its power spectrum, the residuals for each data set and the statistics of the inversion. Further on, its power spectrum is used as a means for inferring the mean magnetic thickness and mean magnetization of the region under study. Finally, we discuss our findings and present conclusions and perspectives of our study.

## 2 Data

The combined use of magnetic field measurements taken at different altitudes requires careful data processing in order to maximize their compatibility (see, e.g., Ravat et al. (2002)). In the followng sections we describe the dedicated processing scheme of each data type included in our joint inversion. All data are selected within a spherical cap of $\theta_0 = 15°$, centered at longitude=22.5° E and latitude=$-25°$ S.





## 2.1 Satellite data

We consider magnetic field CHAMP satellite measurements that were selected and processed following the procedure described by Lesur et al. (2010). The data were rotated into the Solar Magnetic (SM) Cartesian coordinate system (in this coordinate system the Z SM axis coincides with the geomagnetic dipole axis and points to the North and the Y SM axis is perpendicular to the Earth-Sun line and points towards dusk) and then only the X and Y SM components were kept. This was done in order to avoid contamination by the ring current (see Lesur et al. (2008)). In addition, data were selected only for local times between 23:00 and 5:00, when the Z component of the Interplanetary Magnetic Field (IMF) was positive, the norm of the Vector Magnetic Disturbances index (VMD, Thomson and Lesur (2007)) less than 20 nT and the norm of its derivative less than 100 nT/day. From the selected data, the GRIMM lithospheric field model (Lesur et al. (2013)) from SH degree 17 to SH degree 80 was subtracted. This was done in order to avoid a spectral leakage of the lithospheric field into the secular variation model (see, e.g., Lesur et al. (2010)). The residuals were used to construct a time-dependent core field model up to SH degree 18 (with splines of order 6 with a knot spacing of 6 months as the temporal basis function), a static lithospheric field model up to SH degree 30, an external static field model up to SH degree 20 and a time varying external field model based on three different parameterizations. A slowly varying external field component was considered by solving for an axial dipole external term in the GSM coordinate system every 10 years. The more rapidly varying external field components were considered by solving for the SH degree 1 coefficients in the SM coordinates every 100 days. The fact that these bins were 100 days large reduced the risk of leakage due to a correction of a track-by-track type (see Thébault et al. (2012)). Finally the even more rapidly varying external fields were accounted for by solving for the scaling coefficients of the SVMD index, the satellite-based version of the VMD index. This index is obtained by calculating the mean value of the measured magnetic field over each orbit and by subsequently normalizing these values in bins of 100 days (see Kunagu et al. (2013)). To the vector residuals of this modelling procedure the initially removed lithospheric field model was added back. The final CHAMP data set comprises 13229 triplets of vector data, varying between 266 km and 475 km altitude.

The selection and correction procedure of Swarm magnetic field measurements closely follows the one described in Thébault et al. (2016) for mid- and low-latitudes. Magnetic field measurements taken by the lowest pair of the Swarm satellites (Alpha and Charlie) are considered between March 2014 and December 2015. The diurnal ionospheric field contribution is minimized by keeping night time data only (Local Time between 23:00 and 5:00 and sun at least 10 degrees below the horizon). Data taken during disturbed magnetic conditions are rejected by selecting only times for which the Dst index is lower than 5 nT, with a time variation smaller than 5 nT over the three previous hours, and for which the Kp index is lower than 2. The selected measurements are then corrected for independent main (Rother et al. (2013)) and magnetospheric field models (Hamilton (2013)) and are further corrected on a track-by-track basis. The residuals, vector and scalar, are used to construct finite differences between the measurements taken by Swarm Alpha and Charlie. These differences simulate the East-West gradient, which is very efficient in filtering out remaining rapid variations of the external field. The final Swarm dataset contains 27699 triplets of vector Swarm across-track differences and 27690 scalar Swarm across-track differences, varying between 460 and 480 km altitude.





## 2.2 Aeromagnetic data

Since 1970, coordinated efforts have been made by African countries and collaborators towards the aeromagnetic mapping of southern Africa. This African Magnetic Mapping Project (Barritt (1993)) eventually led to the SaNaBoZi aeromagnetic grid, whose name stands for South Africa, Namibia, Botswana and Zimbabwe (Stettler et al. (2000)). New surveys have been

conducted or reprocessed in order to fill the gaps over Angola and Mozambique (Webb (2009)). The SaNaBoZi grid and the grids over Mozambique and Zambia as well as marine data off-shore South Africa have been compiled in the second version of the WDMAM (Dyment et al. (2015), Lesur et al. (2016)) and are used in this study. In order to reduce the data volume and balance the number of near-surface and satellite measurements, the respective grid is resampled to $0.1\,^\circ$, leading to 77626 aeromagnetic and marine scalar measurements.

## 2.3 Magnetic observatory and repeat stations data

Our ground data set comprises magnetic measurements from southern African repeat stations and geomagnetic observatories as processed by Korte and Mandea (2016). From 2005 to 2015 annual repeat station surveys were carried out in the framework of a collaboration between Hermanus Magnetic Observatory (now SANSA Space Sciences) and the German Research Center for Geosciences, GFZ. The surveys comprised 40 stations in South Africa, Namibia and Botswana. Using an on-site variometer

at all stations, the measurements were reduced to (quiet) night time averages (see details in Korte et al. (2007)). Here, we consider the data up to 2009 and use version 3 of the GRIMM model series (Lesur et al. (2010); Mandea et al. (2012)) to eliminate core (up to SH degree 14) and magnetospheric field contributions from the data. The final data set comprises up to five vector magnetic anomaly estimates for each location, in some cases fewer due to gaps in the annual series or the rejection of obvious outlying results. Four INTERMAGNET observatories exist in the southern African region, but as the first available

annual mean value from observatory Keetmanshoop in Namibia dates to 2009.5, measurements from this observatory were not included in the data set. For each one of the other three magnetic observatories, that is Hermanus and Hartebeesthoek in South Africa and Tsumeb in Namibia, nine vector lithospheric anomaly estimates were obtained by subtracting the GRIMM3 core and magnetospheric predictions from the annual mean values. In total, 161 triplets of ground vector measurements over 37 repeat stations and 27 triplets of annual mean values from 3 INTERMAGNET observatories were included in the ground data

set.

The input data set is summarized in Table 1. Additionally, figure 1 shows an histogram of the different data types as a function of the altitude above the Earth's mean reference radius.

## 3  Method

The Revised-Spherical Cap Harmonic Analysis (Thébault et al. (2006)) is used for modeling the lithospheric magnetic field

inside the volume of a spherical cone. The cone is defined as the intersection of two spherical caps defined by two concentric spheres of different radii and an infinite cone of half-angle aperture $\theta_0$, whose apex lies at the Earth's center. In contrast to the



| | vector | scalar | vector across-track differences | scalar across-track differences |
| --- | --- | --- | --- | --- |
| | (number of triplets) | | (number of triplets) | |
| satellite CHAMP | 13229 | | | |
| satellite mission Swarm | | | 27699 | 27690 |
| WDMAM | | 77626 | | |
| magnetic observatories | 27 | | | |
| repeat stations | 161 | | | |

**Table 1.** The number of input data per data type.

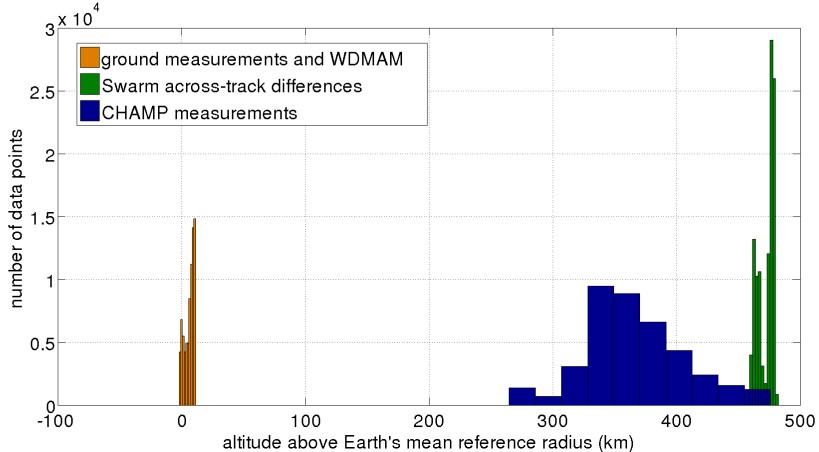

**Figure 1.** The histograms of the different types of magnetic field data included in our model as a function of their altitude above the Earth's mean reference radius (according to the geocentric reference frame).

Spherical Cap Analysis (Haines (1985)), that solves one boundary problem on the lateral boundaries of a spherical cap, the R-SCHA technique solves two boundary problems: one on the lateral boundaries of the spherical cone and one on its upper and lower boundaries. Each one of these two boundary problems generates one set of basis functions, the Legendre and the Mehler basis functions, respectively. The Legendre basis functions are laterally oscillating functions while the Mehler basis functions are radially oscillating and can account for the upward/downward continuation operation of the potential field inside the cone. The magnetic field $\mathbf{B} = \mathbf{B_1} + \mathbf{B_2}$ is the sum of the magnetic field projected on these two basis functions, where $\mathbf{B_1}$ is written as the gradient of a series of the Legendre basis functions

$$\mathbf{B_1}(r,\theta,\phi) = -\nabla \left\{ \alpha \sum_{k\geq 1} \sum_{-k\leq m\leq k} \left[ \left(\frac{\alpha}{r}\right)^{n_k+1} G^{i,m}_{n_k} + \left(\frac{r}{\alpha}\right)^{n_k} G^{e,m}_{n_k} \right] \beta^m_{n_k}(\theta,\phi) \right\}, \tag{1}$$





with

$$
\beta_{n_k}^m(\theta,\phi) = \begin{cases} \cos m\phi\, P_{n_k}^m(\theta), & 0 \le m \le k \\ \sin|m|\phi\, P_{n_k}^m(\theta), & -k \le m \le -1, \end{cases} \tag{2}
$$

where $P_{n_k}^m(\theta)$ are the generalised Legendre functions of integer order $m$ and real degree $n_k$, and $G_{n_k}^{i,m}$ and $G_{n_k}^{e,m}$ the harmonic coefficients of the corresponding order and degree to be estimated via the inverse problem, and $\mathbf{B_2}$ is written as the gradient of a series of the Mehler basis functions

$$
\mathbf{B}_2(r,\theta,\phi) = -\nabla\left\{ \alpha \sum_{p\ge 0} \sum_{-p \le m \le p} R_p(r) G_p^m \gamma_p^m(\theta,\phi) \right\}, \tag{3}
$$

with

$$
R_p(r) = \begin{cases} \sqrt{\dfrac{\alpha}{r}}\left[ \dfrac{2\pi p}{\log\left(\frac{\beta}{\alpha}\right)} \cos\left( p\pi\dfrac{\log\left(\frac{r}{\alpha}\right)}{\log\left(\frac{\beta}{\alpha}\right)} \right) + sin\left( p\pi\dfrac{\log\left(\frac{r}{\alpha}\right)}{\log\left(\frac{\beta}{\alpha}\right)} \right) \right], & p \ge 1 \\[2ex] const, & p = 0 \end{cases} \tag{4}
$$

and

$$
\gamma_p^m(\theta,\phi) = \begin{cases} \cos m\phi\, K_p^m(\theta), & 0 \le m \le p \\ \sin|m|\phi\, K_p^m(\theta), & -p \le m \le -1, \end{cases} \tag{5}
$$

where $K_p^m(\theta)$ are the conical or Mehler functions of integer order $m$ and degree $n_p$, $\alpha$ and $\beta$ are the radii of the lower and upper spherical caps, respectively, and $G_p^m$ are the harmonic coefficients of the corresponding order and degree to be also estimated through the inverse problem.

This problem can be written as a linear system of equations

$$
\mathbf{Gm} = \mathbf{d}, \tag{6}
$$

where $\mathbf{d}$ is the vector of the data, $\mathbf{m}$ is the set of coefficients to be determined, and $\mathbf{G}$ is the matrix of the basis functions. The scalar part of the input data set is linearised through a projection on the ambient field (see, e.g., Blakely (1996)), here taken to be the GRIMM2 core field model centered on the 2005.5 epoch (Lesur et al. (2010)).

The data are inverted in an iteratively re-weighted least square sense with Huber weighting (see, e.g., Farquharson and Oldenburg (1998), Sabaka et al. (2004)), and no regularization is applied. In this case, the coefficients $\mathbf{m}$ are given by

$$
\mathbf{m} = \left(\mathbf{G}^T\mathbf{W}\mathbf{G}\right)^{-1}\mathbf{G}^T\mathbf{W}\mathbf{d}, \tag{7}
$$





where $\mathbf{W}$ is the weighting matrix, which changes at every iteration. It is diagonal and its element $i$ at iteration $j$ is given by

$$w_{ij} = \frac{1}{\sigma_i^2} \min\left(\frac{c\sigma_i}{\|e_{ij}\|}, 1\right), \tag{8}$$

where $\sigma_i$ is the standard deviation of the $i$th point, $e_{ij}$ the residual of the $i$th point at iteration $j$ and $c$ a constant here set to

1.5. Taking into account the expected accuracy, number of points and typical values due to distance from the sources of each

data type, the standard deviation for the ground data is taken to be equal to 20 nT, for the aeromagnetic data equal to 40 nT and

for the satellite data equal to 2 nT. The truncation index for $k$ and $m$ of Eq. 1 is set to 80, which allows for the description of

the field up to 40 km horizontal wavelength. The truncation index for $p$ and $m$ of Eq. 3 is set to 9. This number is constrained

by the availability of data at different altitudes (see paragraph 5.2.2 of Vervelidou (2013) for more details). The radius of the

lower and upper spherical cap was set to -10 km and 500 km, respectively.

In order to estimate the spectral content of the regional model, we compute its R-SCHA surface power spectrum according

to Vervelidou and Thébault (2015), their equation 10:

$$\frac{E_{\vartheta\Omega_\rho}}{S_{\vartheta\Omega_\rho}} = \sum_{m=0}^{\infty} \frac{(1+\delta_{m,0})}{2(1-cos\theta_0)} \sum_{k=0}^{\infty} \left\{ (n_k+1)(2n_k+1)\left(\frac{a}{\rho}\right)^{2n_k+4} \left[\left(G_{n_k}^{i,m}\right)^2 + \left(H_{n_k}^{i,m}\right)^2\right] \right.$$
$$\left. + n_k(2n_k+1)\left(\frac{\rho}{a}\right)^{2n_k-2} \left[\left(G_{n_k}^{e,m}\right)^2 + \left(H_{n_k}^{e,m}\right)^2\right] \right\} \|P_{n_k}^m\|^2, \tag{9}$$

where $\rho$ is the radius of the sphere on the surface of which we calculate the power spectrum, $\vartheta\Omega_\rho$ is that surface and $S_{\vartheta\Omega_\rho}$

its area.

This power spectrum is equivalent to a SH power spectrum (Lowes (1966)), with the only difference that the degrees $n_k$ are

real numbers and depend on the order $m$. Therefore, each term of the double sum of Eq. 9 corresponds to a different degree

$n_k$. In order to be able to associate the terms of this power spectrum to a specific wavelength, we bin together the harmonic

coefficients corresponding to degrees with close numerical values. Each bin is taken to be $\frac{\pi}{\theta_0}$ large, following Vervelidou and

Thébault (2015), and its characteristic wavelength is related to its mean degree $\bar{n}$ according to the known approximation (see

Backus et al. (1996))

$$\bar{\lambda} \approx \frac{2\pi\rho}{\bar{n}+\frac{1}{2}}. \tag{10}$$

## 4 Results

### 4.1 Statistics of the multiscale inversion

The residuals of each data type after the inversion is shown in Figs 2 and 3. Figs 2(a) and (b) show the residuals of the CHAMP

and Swarm data set, respectively. While in the CHAMP data set all three components show satellite tracks being partially or



| | Ground data | | | WDMAM | CHAMP | | | Swarm across-track differences | | | |
|---|---|---|---|---|---|---|---|---|---|---|---|
| | X | Y | Z | | X | Y | Z | X | Y | Z | Scalar |
| **RMS (nT)** | 13.6 | 12.7 | 6.8 | 67.7 | 2.3 | 2.2 | 2.3 | 0.6 | 0.6 | 0.4 | 0.8 |
| **Correlation coefficients** | 0.99 | 0.99 | 1 | 0.8 | 0.7 | 0.77 | 0.84 | 0.85 | 0.73 | 0.95 | 0.83 |

**Table 2.** The residual mean square (RMS) and the correlation coefficient between our model and the input data.

entirely contaminated by external field noise, this is not the case for the residuals of the Swarm differences, and especially so for the Z component.

The residuals between the model and the WDMAM, shown in Figure 2(d), are small scale and reflect the difference between the spatial resolution of our model and the spatial resolution of the input grid. Residuals over the oceanic region are likely due to large gaps in marine magnetic field measurements. These gaps have been filled in with values from the magnetization model of Dyment et al. (2015), which is based on considerations concerning the age of the ocean floor, tectonic motions and geomagnetic polarity reversals.

Figure 3 shows the predictions of our model (red circles) and the mean of the measurements (blue squares) at each repeat station and magnetic observatory considered in our inversion, for X, Y and Z component. The abbreviations used here are those used by Korte and Mandea (2016) (see their Table 3). According to this Figure, our model is in close agreement with the actual measurements. This is a mutual confirmation of both the accuracy of our model and the measurements. The spatial distribution of the residuals is shown in Figure 2(c). According to this figure, the residuals do not exhibit any specific spatial pattern.

The residual mean square (RMS) for each data type and each component are summarized in Table 2. We see that the RMS of all three components of the ground data is below the initial standard deviation, set to 20 nT. This shows that the location of the repeat stations do not lie on very localised anomalies, which is actually one of the objectives when selecting repeat stations locations. The RMS of the Z component is the smallest, which is expected as the horizontal components X and Y are usually more contaminated by external field noise. The RMS of the WDMAM is larger than the initial standard deviation, set to 40 nT. This is because the WDMAM grid has features whose spatial scale are smaller than our model is able to depict. All three components of the CHAMP data are resolved within the initial standard deviation. The Swarm differences show, as expected, smaller misfit values than the CHAMP measurements. The second row of Table 2 shows the correlation coefficient between the model predictions and the data, separately for each data type and component. We see that all data types are well resolved, with correlation coefficients above 0.7. This indicates a good level of compatibility between all different data sets. The lowest correlation is found for the X and Y components of the CHAMP data and the Y component of the Swarm across-track differences (between 0.7 and 0.77), which is expected due to the external field contamination of the horizontal components in satellite measurements, visible also in Figure 2.





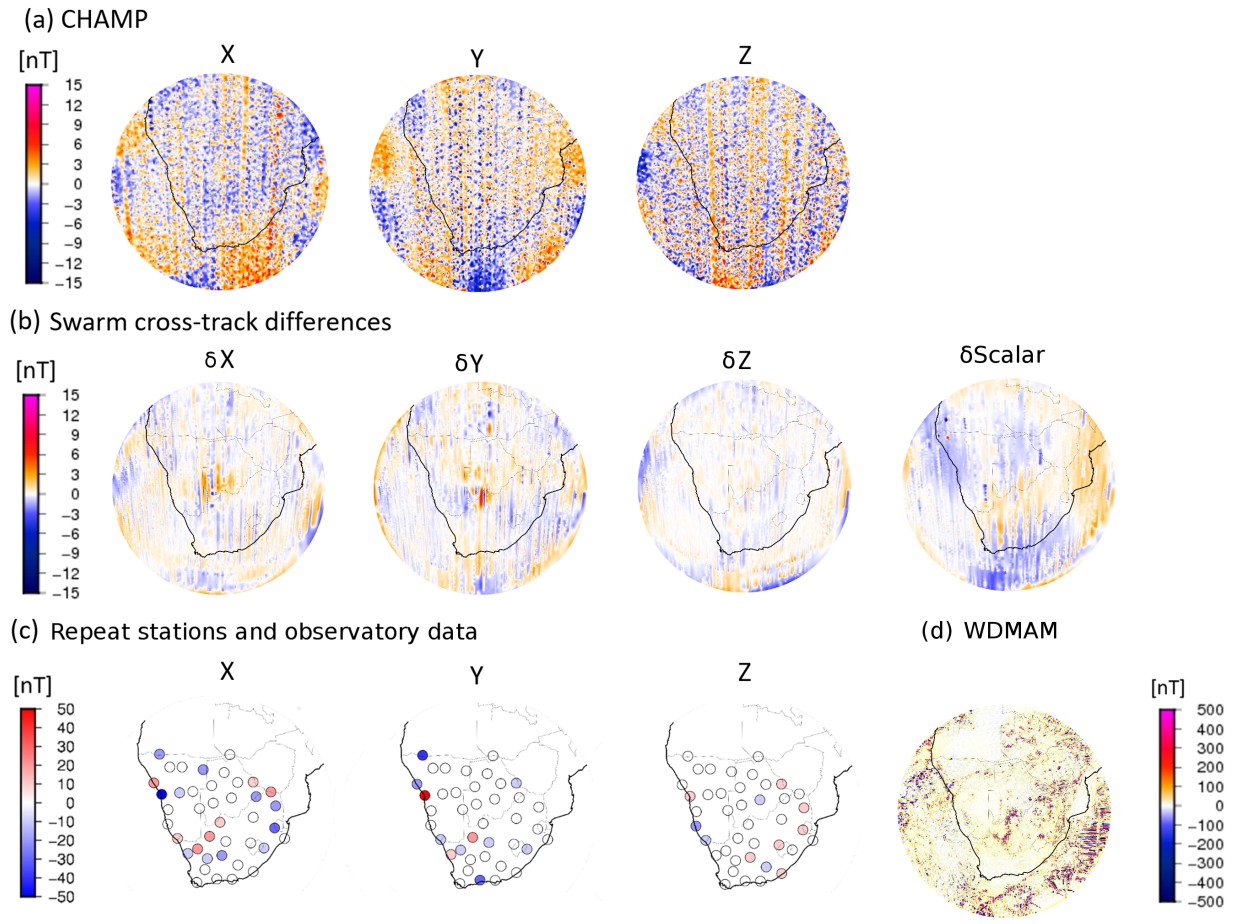

**Figure 2.** Residuals between our model and the input data for (a) the CHAMP vector measurements, (b) the Swarm across-track differences of the vector and scalar measurements, (c) the repeat stations and observatory vector measurements and (d) the WDMAM grid points. Note the different colorscales.



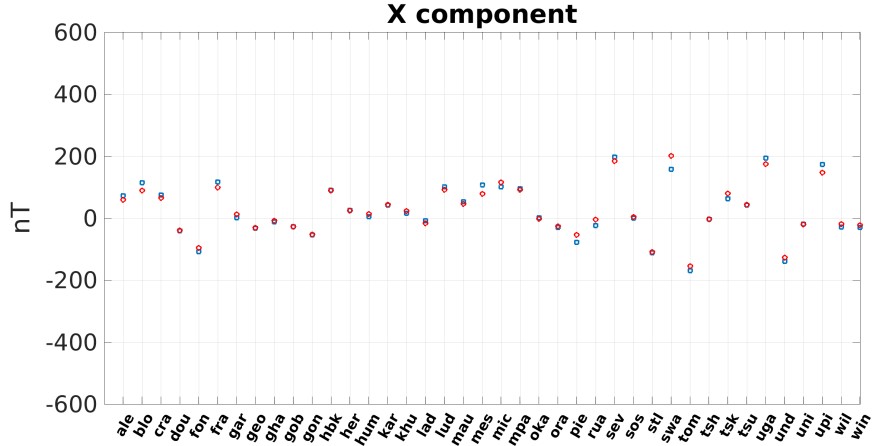

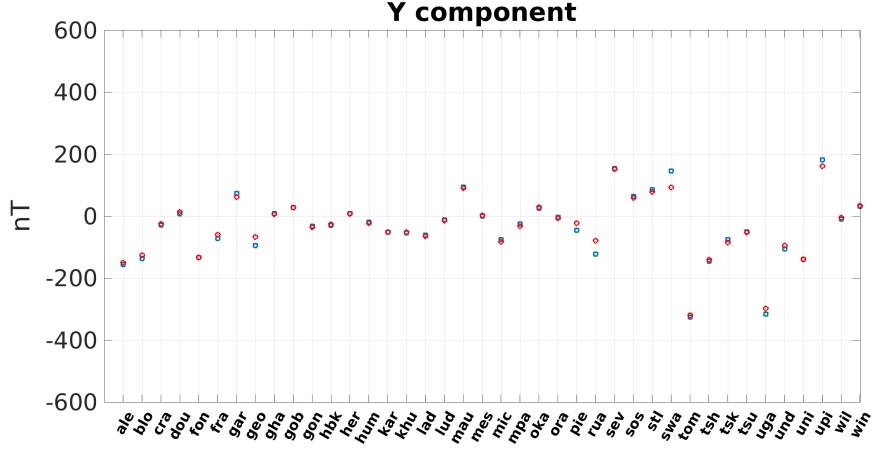

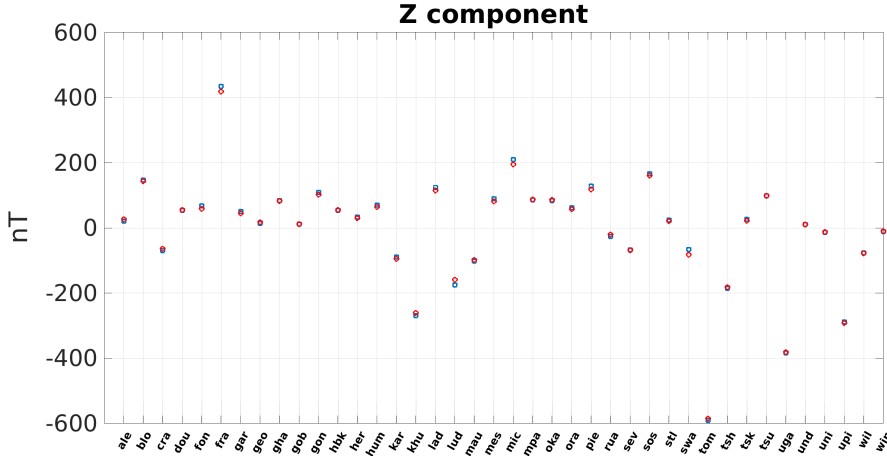

**Figure 3.** The predictions of our model over the ground measurement points (red circles) and the mean of the measurements (blue squares), for the X, Y and Z components. The abbreviations follow Table 3 by Korte and Mandea (2016).




**Figure 4.** (a-c) The X, Y and Z component of our lithospheric field model plotted at the Earth's mean reference radius. (d) Outline of large-scale tectonic features, from Figure 8 by Korte and Mandea (2016).

## 4.2 The model in the spatial and spectral domain

Figure 4 shows the model prediction for X, Y, and Z components at the Earth's mean reference radius. Figure 5 shows the model prediction for the Z component at various altitudes above the Earth's mean reference radius. The different components highlight structures of different orientation and the maps at different altitudes carry information about the lateral and vertical extent of the sources. These maps are discussed in more detail in the Discussion section.





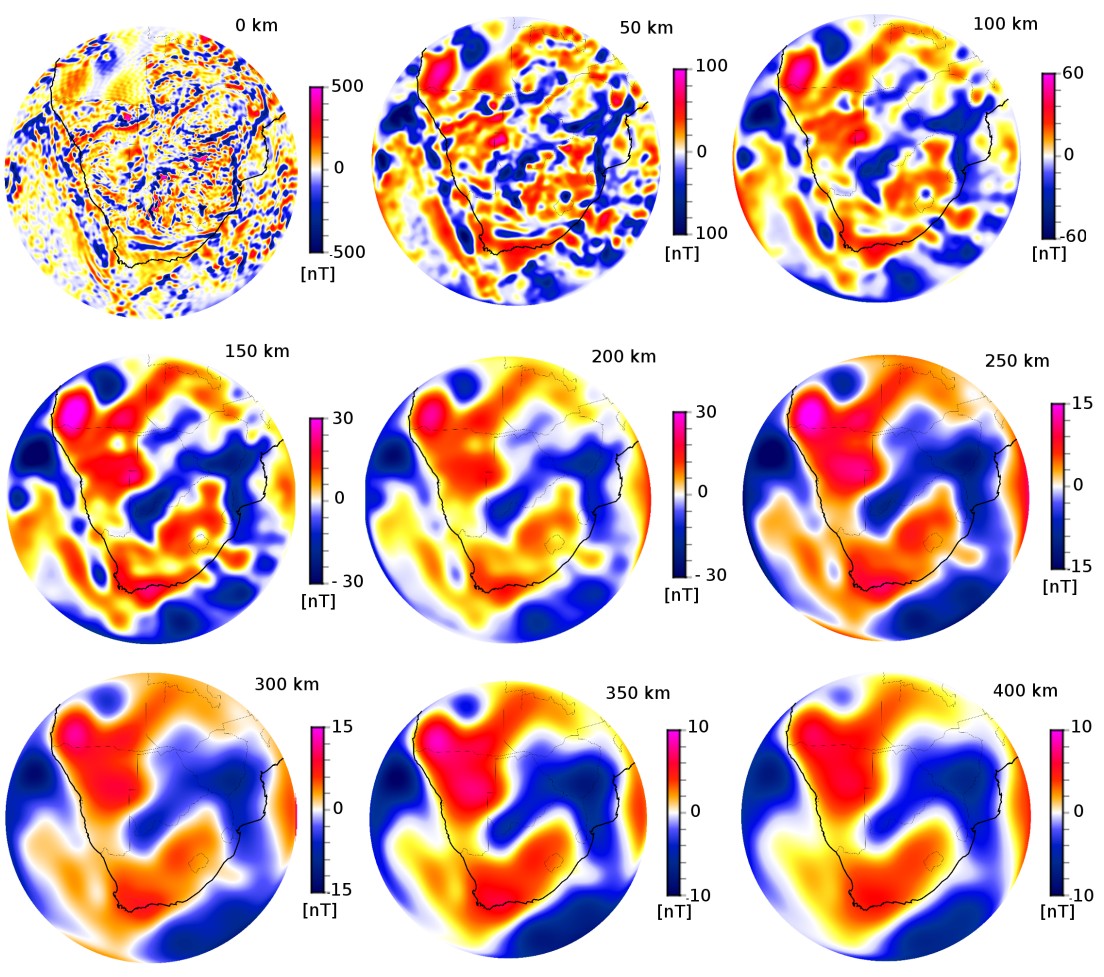

**Figure 5.** The Z component of our lithospheric field model plotted at different altitudes above the Earth's mean reference radius. Note the different colorscales.




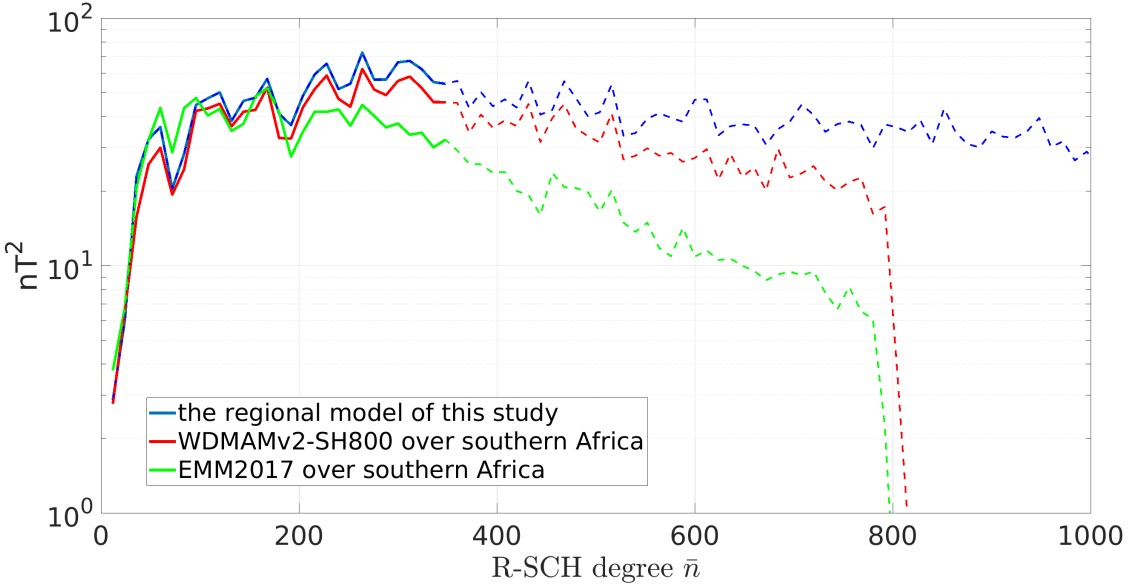

**Figure 6.** The power spectrum of our regional lithospheric field model (blue curve) and the power spectra of the global WDMAMv2-SH800 and EMM2017 models calculated over the region of interest (red and green curves, respectively). Solid line is used for the bandwidth over which the comparison between the spectra of the regional and global models is valid (see text for details).

Figure 6 shows the R-SCHA power spectrum of the model (blue line) up to equivalent SH degree 1000 (spatial wavelength 40 km). It is compared to the power spectra of the WDMAMv2-SH800 model (Lesur et al. (2016)) and the EMM2017 model (see https://www.ngdc.noaa.gov/geomag/EMM/), given in red and green color, respectively. The regional R-SCHA power spectra over southern Africa of these two global models were calculated from their vector predictions at the Earth's mean reference

5    radius within the same spherical cap and for the same inversion parameters used for the derivation of our regional model. Given the choice of $k_{max} = 80$, the comparison is only valid up to degree $\bar{n} = 340$, beyond which the spectral content of global models is not fully recovered (for details see Vervelidou (2013), Chapter 4). To denote the transition limit, we use solid lines for the spectra up to $\bar{n} = 340$ and dashed lines beyond this degree. According to this figure, the power spectrum of the regional model up to SH degree 100, that is the bandwidth constrained by satellite data, lies inbetween the power spectra of

10   WDMAMv2-SH800 and EMM2017. Starting from SH degree 100, the regional model is more energetic than the two global models. As expected, the global models carry no energy beyond degree 800, while the regional model carries energy over a larger bandwidth.

### 4.3 Magnetic thickness and magnetization estimates

We use the power spectrum of our model to estimate the mean magnetic thickness and the mean magnetization over southern

15   Africa. Traditionally, such studies rely on Fourier power spectra (see, e.g., Ravat et al. (2007) and references therein; Rajaram



et al. (2009), Bansal et al. (2011), Salem et al. (2014)). Fourier spectra rely on the flat Earth approximation and are, therefore, valid only on local scales. The R-SCHA power spectrum is defined in spherical coordinates and offers therefore a more accurate description of the energy per wavelength over regional scales. To infer the mean magnetic thickness and magnetization estimates, we fit the observational power spectrum to the statistical power spectrum of the Earth's lithospheric magnetic

field derived by Thébault and Vervelidou (2015). This statistical expression is based on the following three assumptions: the thickness of the magnetized shell is finite and constant, the inducing field is axial dipolar, and the power spectrum of the susceptibility follows a power law. This expression has been shown to fit well high resolution lithospheric field models (see Vervelidou (2013), section 4.3; Thébault and Vervelidou (2015), their Figure 4; Vervelidou and Thébault (2015), their Figure 5).

The misfit functional between the observational and the statistical power spectra is defined as follows (Voorhies et al. (2002), equation 14a)

$$\chi^2 = \sum_n \left[ ln\left(R_n\right) - ln\left(E\{R_n\}\right) \right]^2, \tag{11}$$

where $R_n$ is the observational power spectrum and $E\{R_n\}$ its expected value according to the statistical expression of Thébault and Vervelidou (2015) (their Eq. 26), which depends on three parameters: the thickness of the magnetized layer, the

magnetization intensity and the power law that characterises the power spectrum of the susceptibility distribution. The co-estimation of the three unknown parameters is unstable (see, e.g., Bouligand et al. (2009)). A possible way to alleviate some of the ambiguity is to rely on a priori information about the susceptibility power law. Here, we set its value to 1.36, in agreement with the global estimate of Thébault and Vervelidou (2015), based on the susceptibility map of Hemant and Maus (2005) and the NGDC-720 lithospheric field model (Maus (2010)).

We calculate the misfit functional $\chi^2$ for a variety of different thickness and magnetization values. This provides us with the 2D function, shown in Figure 7(a). As expected, there is a trade-off between the magnetization and the magnetic thickness. We find that Eq. 11 has its minimum for an equivalent magnetization equal to 0.7 A/m, which is twice as large as the global average estimated by Thébault and Vervelidou (2015), and for a magnetic thickness equal to 16 km. Considering the misfit values up to 3 standard deviations above the misfit minimum, we find that the mean magnetic thickness lies between 11 and 22 km and the

mean magnetization between 0.6 and 0.9 A/m (the respected values are marked with black color in Figure 7(a)). This estimate is in close agreement with the 20 km mean magnetic thickness proposed by Maus et al. (1997) following a similar spectral analysis but based on a cartesian statistical expression for the power spectrum of the Earth's lithospheric magnetic field and a Fourier observational power spectrum over southern Africa given by Whaler (1994).

## 5  Discussion

The regional field model derived in this study offers insights into several aspects of lithospheric field modeling. The first concerns the contamination of lithospheric field models by external field sources (see Finlay et al. (2017) and Thébault et al.





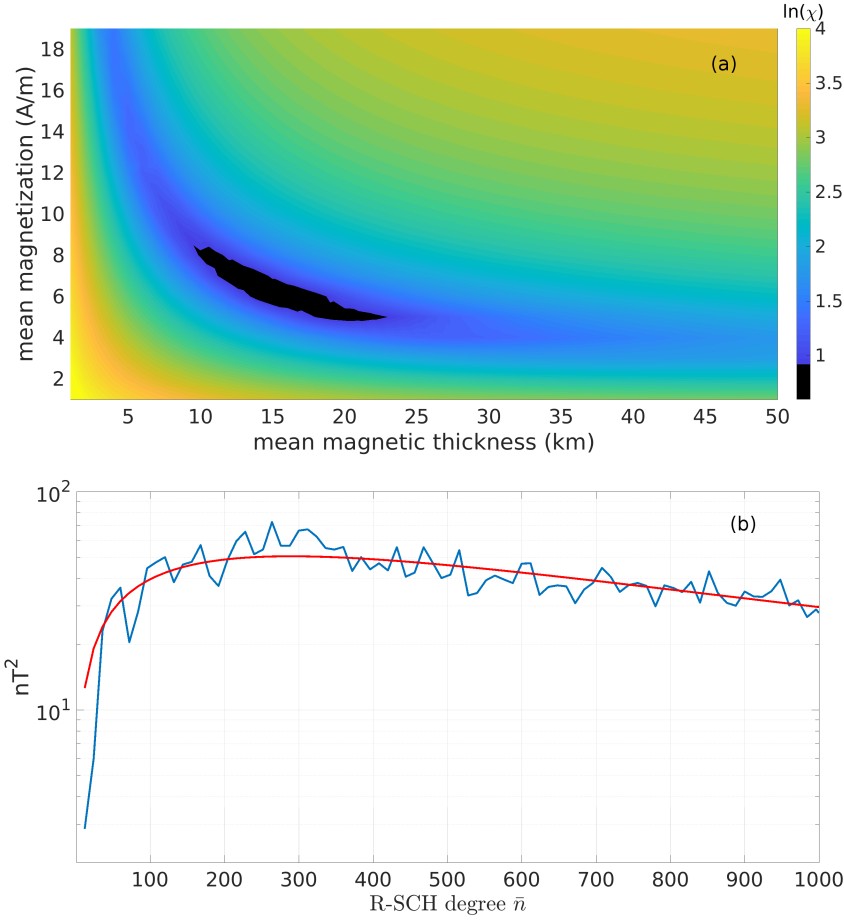

**Figure 7.** (a) The natural logarithm of the misfit (Eq. 11) between the R-SCHA power spectrum of our model and the statistical expression of Thébault and Vervelidou (2015) for a range of values for the mean magnetic thickness and magnetization. Values within the 3-$\sigma$ confidence interval are marked with black color. (b) The R-SCHA power spectrum of our model (blue curve) and the statistical expression for 16 km magnetic thickness and 0.7 A/m magnetization (red curve).



(2017) for reviews on this topic). One major source of contamination is the magnetic field generated by the magnetospheric ring current. The strength of this current fluctuates rapidly under the effect of the solar wind even during magnetically quiet times and so does the associated magnetic field. A single satellite cannot distinghuish between these time variations and the small spatial scale variations of the lithospheric field, because of its constantly moving position. As this signal of magnetospheric

origin changes from one track to another (due to this signal's temporal variation), it introduces offsets between adjacent satellite tracks. This effect is illustrated by the residuals between the model and the satellite CHAMP measurements shown in Fig. 2(a). Figure 2(b), depicting the residuals between the model and the Swarm measurements, does not show similar prominent features. Although the pattern of the residuals is not random, especially for components X and Y, the particular effect of track offsets is mainly eliminated. This is due to the configuration of the Swarm mission, and in particular due to its two satellites, Alpha

and Charlie, that fly side by side. Given the large scale geometry and the longitudinal symmetry of the ring current's magnetic effect at LEO altitude, taking the across-track differences of Alpha and Charlie measurements is an efficient mean to reduce its effect in lithospheric field models.

The second aspect concerns the flexibility of regional modeling in the use of regularization. As shown in Fig. 6, the regional model carries more energy than the global ones over the bandwidth constrained by near-surface measurements. Given that these

are scalar values, the global models WDMAMv2-SH800 and EMM2017 were derived under some regularization constraints (Lesur et al. (2016), Chulliat (2017)) in order to deal with the Backus effect, whose signature is particularly prominent around the equator (Backus (1970)). Our regional model does not exhibit such artifacts, although we did not apply any regularization. We conclude from the above that regional modeling offers the flexibility of applying regularization that is appropriate to the region of interest instead of a generic regularization that affects the model globally while seeking to smooth out small scale

local artifacts.

The third aspect concerns the estimation of the magnetic thickness and magnetization through spectral analysis of lithospheric field models. Thébault and Vervelidou (2015) noted that the information about the magnetic thickness lies principally in the SH degree where the peak of the power spectrum occurs. Moreover, that this generally falls within the bandwidth known as the spectral gap to denote that it corresponds to spatial scales poorly constrained both by satellites and near-surface

measurements. In addition to this uncertainty, independent estimates of the magnetic thickness and the magnetization require lithospheric field models of very high resolution. As already noted in Section 4.2, our model resolves fully spatial scales up to SH degree 340. Pushing this limit to higher degrees and/or closing down the spectral gap, will eventually modify the shape of the power spectrum and therefore our magnetization and magnetic thickness estimates. Here we found that the 3-$\sigma$ confidence interval for the magnetization and magnetic thickness is [0.6, 0.9] A/m and [11, 22] km, respectively. The estimate of

the magnetization is two to three times larger than the global average as estimated by Thébault and Vervelidou (2015), while the magnetic thickness lies below the global average. Considering the cratonic nature of the southern african crust, this result might reflect an inadequate separation between the magnetization and the magnetic thickness.

Finally, the high resolution vectorial maps of the magnetic field model displayed at different altitudes offer important insights concerning the sources of the observed anomalies. Although a thorough description and interpretation of these maps is beyond

the scope of our study, we point out some striking features of the regional lithospheric field. Some of the prominent anomalies



visible in the X component, shown in Figure 4, include the East-West anomaly along the Damara Belt in Namibia, the Beattie anomaly in South Africa and the magnetic anomaly along the Limpopo belt, that lies between the Zimbabwe and the Kaapvaal Craton. The first two appear to be relatively shallow features because they are not detected above 50 km altitude (see Figure 5). However, the anomaly along the Limpopo belt persists at higher altitudes suggesting a deeper and/or elongated origin of the

corresponding magnetized structure. The Y component brings forth oceanic magnetic lineations along the west coast of South Africa and Namibia and prominent anomalies along the east borders of the Kaapvaal Craton and into the Mozambique Belt. These oceanic magnetic lineations are still visible up to 150 km altitude. We infer from this that their source lies in the lower crust. The anomaly over Mozambique beyond 50 km altitude merges with the anomaly over the Limpopo belt. As expected, all these anomalies are also visible in the Z component. A strong anomaly, visible in all three components, is a north-south

structure in the north-central part of South Africa. We argue that the source of this anomaly is near-surface and small scale, firstly because this anomaly is clearly seen in the residuals of the WDMAM grid (see Figure 2(d)) and secondly because looking at Figure 5, we observe that it is only visible up to 50 km altitude, and then it merges with anomalies of deep origin, possibly related to the Limpopo belt. Its location coincides with an important Band Iron Formation (see, e.g., Klein and Beukes (1989)).

## 15  6   Conclusions and perspectives

In this study we modelled jointly satellite, aeromagnetic, marine and ground magnetic field measurements and derived a lithospheric magnetic field model over southern Africa. We showed that all different data sets are highly compatible with each other, with the horizontal satellite components being the least compatible with the rest of the data set due to external field contamination. We showed that Swarm across-track differences at mid-latitudes eliminate efficiently the offsets between

adjacent satellite tracks due to the rapid variations of the ring current (see Fig. 2). Mutual exchange of expertise among specialists in internal and external magnetic field sources will help address further the issue of external field contamination of internal field models (see Stolle et al. (2018) for a relevant compilation of articles).

By means of the R-SCHA power spectrum we showed that our model carries more energy over the bandwidth constrained by near-surface measurements than global high resolution lithospheric field models over the same region, as it does not require

regularization (see Fig. 6). The extension of our work to a global model through successive regional analyses is straightforward (see Thébault (2006) and Thébault et al. (2016) for such studies based on satellite data). Thanks to the R-SCHA power spectrum, the spectral content of such a global model can be evaluated regionally. This is crucial because contrary to satellite data that have an almost homogeneous data coverage, grids based on near-surface data are extremely inhomogeneous in terms of data quality, availability and spatial resolution (see Vervelidou and Thébault (2015) for a regional spectral analysis of the

NGDC-720 model).

A wealth of information about the sources of lithospheric field anomalies can be extracted from our regional, high resolution, vector lithospheric field model that can be accurately upward and downward continued (see Figures 4 and 5). A comprehensive

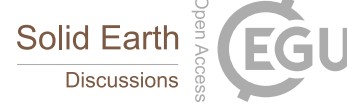

interpretation of these anomalies will benefit from a joint analysis of different geophysical data sets, like gravity, heat flux and seismic data (see Webb (2009) for a compilation of such data sets over southern Africa).

Through a spectral analysis of our model we inferred that the mean magnetic thickness and mean magnetization over southern Africa lies between 11 and 22 km and 0.6 and 0.9 A/m, respectively. More accurate mean estimates can be obtained by

5 means of lithospheric models of higher spatial resolution and more realistic expressions of the power spectrum of the Earth's lithospheric magnetic field. In addition, new spectral tools defined on spherical coordinates will help bridge the gap between Fourier power spectra, that have the ability to represent the energy over small regions but rely on the flat Earth approximation, and the R-SCHA power spectra that provide average estimates over large regions.

*Data availability.* Grids of our lithospheric field model at various altitudes are available in the supplementary material.

10 *Competing interests.* We declare that no competing interests are present.

*Acknowledgements.* The authors wish to thank Vincent Lesur for making available the processed CHAMP satellite data set and the South African National Space Agency (SANSA) Space Science in Hermanus for the collaborative effort to obtain and process the southern African repeat station data. FV was partly funded by Région Île-de-France and partly by the Deutsche Forschungsgemeinschaft (DFG, German Research Foundation), within the Schwerpunktprogramm 1788-Dynamic Earth under the grant LE2477/7-1.



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
