# Peer review of "A high resolution lithospheric magnetic field model over southern Africa based on a joint inversion of CHAMP, Swarm, WDMAM and ground magnetic field data"

_Solid Earth, 2018_

## Referee Comment (RC1) · Anonymous Referee #1 · 3 May 2018

The manuscript "A high resolution lithospheric magnetic field model over southern Africa based on a joint inversion of CHAMP, Swarm, WDMAM and ground magnetic field data" was submitted by three highly known scientists: Foteini Vervelidou, Erwan Thébault and Monika Korte. The overall impression after reading this manuscript The paper addresses relevant scientific topics and fall in the scope of SE. An inversion of different magnetic field sources (CHAMP, SWARM, WDMAM and three South African observatories) is described and applied to derive the field of lithospheric field. After an introduction and outline of motivation the used method and mathematical formalism

are carefully explained. For used inversion scheme the "revised spherical cap har-monics" have been applied to handle measurements which were observed at different heights. The quality of the different data sets and the methodology itself are explained in an understandable way. The general impression after reading the manuscript is pos-itive; the styling in which the manuscript was written corresponds to a good language level - as far as I can judge this as a non-native speaker. The overall presentation is well structured and clear. Figs. 4a – d present the results and show that the (new) lithospheric field, calculated from the mentioned sources provide new insights in the magnetic properties of the rocks in South Africa. The geological-tectonic interpretation and also the discussion of the rock parameters of the earth are reduced in favor of an assessment of the performance and quality of the mathematical algorithms. This is understandable from the authors' view point, who are more physicists than geologists. But in this direction it would have been interesting to learn more about the interpreta-tion of the new acquired field and how far it can reflect the magnetic effect of previously unknown structures. The number and quality of references are appropriate and the de-scription of experiments and calculations are sufficiently complete and precise to allow their reproduction by fellow scientists at any time.

---

## Referee Comment (RC2) · D. Ravat (Referee) · 14 May 2018

[revised manuscript text omitted]
_0 = 15\,^\circ$, centered at longitude=22.5 $^\circ$ E and latitude=$-25\,^\circ$ S.

[Figure]

[Figure]

**2.1 Satellite data**

is X SM toward the Sun?

We consider magnetic field CHAMP satellite measurements that were selected and processed following the procedure described by Lesur et al. (2010). The data were rotated into the Solar Magnetic (SM) Cartesian coordinate system (in this coordinate system the Z SM axis coincides with the geomagnetic dipole axis and points to the North and the Y SM axis is perpendicular to the Earth-Sun line and points towards dusk) and then only the X and Y SM components were kept. This was done in order to avoid contamination by the ring current (see Lesur et al. (2008)). In addition, data were selected only for local times between 23:00 and 5:00, when the Z component of the Interplanetary Magnetic Field (IMF) was positive, the norm of the Vector Magnetic Disturbances index (VMD, Thomson and Lesur (2007)) less than 20 nT and the norm of its derivative less than 100 nT/day. From the selected data, the GRIMM lithospheric field model (Lesur et al. (2013)) from SH degree 17 to SH degree 80 was subtracted. This was done in order to avoid a spectral leakage of the lithospheric field into the secular variation model (see, e.g., Lesur et al. (2010)). The residuals were used to construct a time-dependent core field model up to SH degree 18 (with splines of order 6 with a knot spacing of 6 months as the temporal basis function), a static lithospheric field model up to SH degree 30, an external static field model up to SH degree 20 and a time varying external field model based on three different parameterizations. A slowly varying external field component was considered by solving for an axial dipole external term in the GSM coordinate system every 10 years. The more rapidly varying external field components were considered by solving for the SH degree 1 coefficients in the SM coordinates every 100 days. The fact that these bins were 100 days large reduced the risk of leakage due to a correction of a track-by-track type (see Thébault et al. (2012)). Finally the even more rapidly varying external fields were accounted for by solving for the scaling coefficients of the SVMD index, the satellite-based version of the VMD index. This index is obtained by calculating the mean value of the measured magnetic field over each orbit and by subsequently normalizing these values in bins of 100 days (see Kunagu et al. (2013)). To the vector residuals of this modelling procedure the initially removed lithospheric field model was added back. The final CHAMP data set comprises 13229 triplets of vector data, varying between 266 km and 475 km altitude.

The selection and correction procedure of Swarm magnetic field measurements closely follows the one described in Thébault et al. (2016) for mid- and low-latitudes. Magnetic field measurements taken by the lowest pair of the Swarm satellites (Alpha and Charlie) are considered between March 2014 and December 2015. The diurnal ionospheric field contribution is minimized by keeping night time data only (Local Time between 23:00 and 5:00 and sun at least 10 degrees below the horizon). Data taken during disturbed magnetic conditions are rejected by selecting only times for which the Dst index is lower than 5 nT, with a time variation smaller than 5 nT over the three previous hours, and for which the Kp index is lower than 2. The selected measurements are then corrected for independent main (Rother et al. (2013)) and magnetospheric field models (Hamilton (2013)) and are further corrected on a track-by-track basis. The residuals, vector and scalar, are used to construct finite differences between the measurements taken by Swarm Alpha and Charlie. These differences simulate the East-West gradient, which is very efficient in filtering out remaining rapid variations of the external field. The final Swarm dataset contains 27699 triplets of vector Swarm across-track differences and 27690 scalar Swarm across-track differences, varying between 460 and 480 km altitude.

Also, in conjunction with CHAMP and Swarm along track gradient, better constraining the overall variation of lithospheric fields (Olsen et al., 2017)....at least at the present high elevations of Swarm

[revised manuscript text omitted]

which leads to close to 95% efficiency in converging when the data are normally distributed (references)

give reason for 1.5

where $\sigma_i$ is the standard deviation of the $i$th point, $e_{ij}$ the residual of the $i$th point at iteration $j$ and $c$ a constant here set to
1.5. Taking into account the expected accuracy, number of points and typical values due to distance from the sources of each
5  data type, the standard deviation for the ground data is taken to be equal to 20 nT, for the aeromagnetic data equal to 40 nT and
*for all components* for the satellite data equal to 2 nT. The truncation index for $k$ and $m$ of Eq. 1 is set to 80, which allows for the description of
the field up to 40 km horizontal wavelength. The truncation index for $p$ and $m$ of Eq. 3 is set to 9. This number is constrained
by the availability of data at different altitudes (see paragraph 5.2.2 of Vervelidou (2013) for more details). The radius of the
lower and upper spherical cap was set to -10 km and 500 km, respectively.

-10 km? shouldn't the boundary of the spherical cap be at the minimum data level?

10  In order to estimate the spectral content of the regional model, we compute its R-SCHA surface power spectrum according
to Vervelidou and Thébault (2015), their equation 10:

$$\frac{E_{\vartheta\Omega_\rho}}{S_{\vartheta\Omega_\rho}} = \sum_{m=0}^{\infty} \frac{(1+\delta_{m,0})}{2(1-cos\theta_0)} \sum_{k=0}^{\infty} \left\{ (n_k+1)(2n_k+1) \left(\frac{a}{\rho}\right)^{2n_k+4} \left[ \left(G_{n_k}^{i,m}\right)^2 + \left(H_{n_k}^{i,m}\right)^2 \right] \right.$$
$$\left. + n_k(2n_k+1) \left(\frac{\rho}{a}\right)^{2n_k-2} \left[ \left(G_{n_k}^{e,m}\right)^2 + \left(H_{n_k}^{e,m}\right)^2 \right] \right\} \|P_{n_k}^m\|^2, \tag{9}$$

where $\rho$ is the radius of the sphere on the surface of which we calculate the power spectrum, $\vartheta\Omega_\rho$ is that surface and $S_{\vartheta\Omega_\rho}$
15  its area.

This power spectrum is equivalent to a SH power spectrum (Lowes (1966)), with the only difference that the degrees $n_k$ are
real numbers and depend on the order $m$. Therefore, each term of the double sum of Eq. 9 corresponds to a different degree
$n_k$. In order to be able to associate the terms of this power spectrum to a specific wavelength, we bin together the harmonic
coefficients corresponding to degrees with close numerical values. Each bin is taken to be $\frac{\pi}{\theta_0}$ large, following Vervelidou and
20  Thébault (2015), and its characteristic wavelength is related to its mean degree $\bar{n}$ according to the known approximation (see
Backus et al. (1996))

$$\bar{\lambda} \approx \frac{2\pi\rho}{\bar{n} + \frac{1}{2}}. \tag{10}$$

**4   Results**

**4.1   Statistics of the multiscale inversion**

25  The residuals of each data type after the inversion is shown in Figs 2 and 3. Figs 2(a) and (b) show the residuals of the CHAMP
and Swarm data set, respectively. While in the CHAMP data set all three components show satellite tracks being partially or

[Figure]

*CHAMP high RMS for residuals in also because of its lower altitude and greater signal magnitude*

| | Ground data | | | WDMAM | CHAMP | | | Swarm across-track differences | | | |
|---|---|---|---|---|---|---|---|---|---|---|---|
| | X | Y | Z | | X | Y | Z | X | Y | Z | Scalar |
| RMS (nT) | 13.6 | 12.7 | 6.8 | 67.7 | 2.3 | 2.2 | 2.3 | 0.6 | 0.6 | 0.4 | 0.8 |
| Correlation coefficients | 0.99 | 0.99 | 1 | 0.8 | 0.7 | 0.77 | 0.84 | 0.85 | 0.73 | 0.95 | 0.83 |

**Table 2.** The residual mean square (RMS) and the correlation coefficient between our model and the input data.

entirely contaminated by external field noise, this is not the case for the residuals of the Swarm differences, and especially so for the Z component.

*relatively*

The residuals between the model and the WDMAM, shown in Figure 2(d), are small scale and reflect the difference between the spatial resolution of our model and the spatial resolution of the input grid. Residuals over the oceanic region are likely

5   due to large gaps in marine magnetic field measurements. These gaps have been filled in with values from the magnetization model of Dyment et al. (2015), which is based on considerations concerning the age of the ocean floor, tectonic motions and geomagnetic polarity reversals. *It would have been good to also use a model based on oceanic data alone, rather than oceanic magnetization model, but perhaps for this area, the oceanic model is sufficiently good...*

[revised manuscript text omitted]

---

## Author Comment (AC1) · 19 Jun 2018

[Figure]

Figure 1: The X, Y and Z component of the residuals between our lithospheric field model and the EMM2017 model, plotted at the Earth's mean reference radius.

[Figure]

Figure 2: The X, Y and Z component of the residuals between our lithospheric field model and the WDMAMv2-SH800 model, plotted at the Earth's mean reference radius.

**Reply to RC1**

We would like to thank the reviewer for the encouraging comments.

Concerning the comment "But in this direction it would have been interesting to learn more about the interpretation of the new acquired field and how far it can reflect the magnetic effect of previously unknown structures." we would like to reply that as far as we can judge our model does not reflect previously unknown structures but does delineate more clearly structures already visible in existing models. In this respect, we present here the residuals between our model and the EMM2017 model (Figure 1) and the residuals between our model and the WDMAMv2-SH800 model (Figure 2).

Figure 1 is dominated by medium-scale patterns, which reflects the fact that our model carries more energy than the EMM2017 model over a large bandwidth. This is apparent already from Figure 6 of the manuscript. Figure 2 shows much smaller-scale features, as expected again based on Figure 6 of the manuscript. But even though the difference between our model and the WDMAMv2-SH800 is principally small-scale, our model describes in more detail several structures, e.g., the magnetic signature of the BIF over Botswana, the Damara Belt in Namibia and the southern border of the Zimbabwe

craton.

Keeping in mind that all three are just models, each with its own restrictions and uncertainties, we do not wish to present and discuss these residuals in the manuscript. We consider that our main point is sufficiently illustrated through Figure 6: our model does not dilute the magnetic signal the way the global models do because we do not need to regularize our solution.

---

## Author Comment (AC2) · 19 Jun 2018

**Reply to RC2**

We would like to thank the reviewer, Dhananjay Ravat, for his detailed feedback and helpful suggestions.

In the following we reply to the reviewer's comments and present the respective modifications made to the manuscript. All lines refer to the initially submitted version.

- Page 1 lines 11, 21 and page 17 line 26

  Following his advice, we rephrased all sentences started with "Thanks to".

- Page 1, first paragraph

  We implemented the suggestions.

- Page 2, line 1

  We corrected the reference Korhonen et al., 2007.

- Page 2, line 23

  We added at the end of the sentence "for modeling the lithospheric magnetic field."

- Page 2, line 28

  Corrected.

- Page 3, concerning the question about the X component of SM coordinates

  We added the phrase "the Sun-Earth line lies on the xz plane" (so, yes, the X SM points towards the Sun) and added the reference of Laundal and Richmond, 2017, where different coordinate systems are explained.

- Page 3, lines 31-32

  We do not use along-track gradients, neither of CHAMP nor of Swarm. We use vector magnetic field measurements of CHAMP and across-track differences of Swarm.

  In line 31 we added the phrase: (hereafter called "across-track differences").

  We added the reference of Kotsiaros and Olsen, 2012 in line 32.

- Page 4, line 19

  We changed "outlying results" to "outliers", as suggested.

- Page 5, Table 1

  We have used differences between the measurements (vector and scalar) of Swarm Alpha and Swarm Charlie. We call these differences "across-track differences". We introduce now this term in page 3, line 31.

- Page 7, line 4

  Concerning the choice of the constant of the Huber weighting we added "following Sabaka et al., 2015".

- Page 7, line 6

  Yes, the value of 2 nT applies to all components. We clarify this now both for the ground data and the satellite data.

- Page 7, line 8-9

  The lower and upper surfaces are taken slightly below and slightly above, respectively, the data level. We added the phrase "in order to avoid artifacts due to edge effects".

- Page 7, line 26

  Corrected.

- Page 8, line 3

  Here we do not wish to say that the amplitude of the residuals is small but rather that the residuals are comprised of short wavelengths. We rephrased as follows: "The residuals...are of short wavelengths and reflect..".

- Page 8, line 5-7

  The gaps at the eastern off-shore region are particularly large (see Figure 2, Lesur et al., 2016). Leaving these regions void of data would create much probably important artifacts.

- Page 11, Figure 4

  The small scale oscillations observed over the north-western part of our maps are caused by the contrast between regions with good and poor near-surface measurements coverage. In particular, as pointed out by the reviewer, no near-surface measurements are available over Angola and some of the adjacent regions. Unfortunately, Gibbs effect cannot be completely avoided through regularization. On the contrary, regularization would cause an artificial smoothing of genuine structures where the near-surface information is available and would jeopardize the subsequent quantitative estimate of the model's power spectrum by artificially changing its slope.

  To better guide the reader and to avoid a possible overinterpretation of the affected structures, we now cover the respective region in grey shade in Figures 4 and 5 and we explain shortly in the manuscript this problem by stating after the first two sentences of paragraph 4.2 the following:

"The results over Angola and adjacent regions, up to 50 km altitude, are shaded with grey because WDMAM does not include there near-surface data and is filled in only with the values of the satellite-based model GRIMM_L120 (Lesur et al. (2013), Lesur et al. (2016)). Such a contrast in the spectral content of the available magnetic field measurements introduces artificial small-scale oscillations, an effect known as ringing. In order to avoid a possible overinterpretation of these structures we exclude them from our model."

- Page 13, line 9

  Corrected.

- Page 14, line 8

  Corrected.

- Page 16, line 4

  Corrected.

- Page 17, lines 5-8

  We think these particular structures are better outlined looking at the Y component rather than at the Z.

  We erased the sentence about their origin.

---

## Author Comment (AC4) · 19 Jun 2018

Please find attached the revised manuscript. Changes are not tracked.

Please also note the supplement to this comment:
https://www.solid-earth-discuss.net/se-2018-26/se-2018-26-AC4-supplement.pdf

———————————————————